# The discourse of civic pride: Hawleri identity as one of the oldest Kurdistani cities in the Middle East

**Kawa Abdulkareem Sherwani**⬛*

Media Techniques Department, Erbil Technical Administrative College, Erbil Polytechnic University, Erbil, Iraq

* kawa@epu.edu.iq

**Data Availability Statement:** All relevant data are within the manuscript.

**Funding:** The author received no specific funding for this work.

## Abstract

Local identity and civic pride have not been comprehensively taken into consideration as the main parameters in the previous studies related to discourse and identity, especially in most of the developing countries. In other words, discourse analysts have not thoroughly studied the mentioned parameters, and systematic data on this path are very scarce. For that purpose, a critical discourse analysis approach was used to study the city identity of "Hawleri" people of Erbil city which is the capital and the most populated city in the Iraqi Kurdistan Region, and known as a center for the worship of the Mesopotamian goddess Ishtar. Thus, the ultimate goals of this study are to first understand how urban residents tend to group themselves according to the cities and communities they live in, and then to show how they proudly affiliate themselves to geographical regions. The data are taken from the city social media and through a survey distributed among people in Erbil. In order to achieve the goals of this study, the author attempted to investigate (i) how civic pride and urban identity are formed, (ii) in what ways people try to group themselves in the cities, (iii) what is the role of culture in shaping the community identity, (iv) who is called Hawleri, and (v) does the language variety have an impact on speakers' civic identity, through studying the place, experience, emotions, history, culture, beliefs and language variety of Hawleri people. Additionally, the total number of participants is 809 people (236 people from the online community and 573 people from the survey). This study concludes that civic pride and city identity are found in the discourse of most people. Hawleri people, as the residents of the oldest city in the Middle East, tend to show this feeling and belonging through speaking a local variety of Kurdish language, their textiles and their common culture, history and geographical birthplace. These sentimentalities sometimes lead to discrimination, bias and racism among different ethnicities living in the city.

## 1. Introduction

Studying the identity of urban people is not easy due to the fact that apart from identity and discourse, this kind of study interferes with a lot of other disciplines such as sociology, politics,

**Competing interests:** The authors have declared that no competing interests exist.

psychology, and research methodology in urban studies. The focus of this study is mainly on the discourse of civic pride and city identity due to the facts mentioned in the following paragraphs.

In studying the identity of any city, many parameters need to be taken into consideration. The most important one is civic pride. Mumford [1] believed that civic pride was an essential part of the history of urban life. The cities constitute unique communities where people have a common purpose and a sense of common identity. The mentioned author stated that "From the Athenian polis, to the Italian city-states, to the cities of the industrial revolution, to the post-industrial cities of today, civic pride has represented a key-value and aspiration of local government, bound up in notions of self-determination, cultural identity, citizenship and belonging."

Moreover, people of different cities have tried to compete with each other in order to save their identities and to show that they are different. Harvey [2] pointed out that rivalry and competition between places have also a role in shaping Civic pride and local identity. In fact, these rivalries and competition between the cities may attract the attention of political communities and local governments. Accordingly, most of the political's parties have been tried to invest this sense of affiliation and antagonism to rebuild civic pride [3, 4].

City identity and place affiliation are multidimensional in the sense of a feeling or belief of inclusion. The shared identity of the diverse inhabitants of a place can be based on a shared place, culture, history, experience, emotion, language, culture, etc. They try to learn from each other and share their values and beliefs with each other. As reported by Vine [5], events have a key role to bring people together so that they learn with and from each other. Through this learning and sharing in active citizenship, a core of shared civic values can be developed.

Place has an important role in shaping people's identity. Urban identity is a new term that often has been used by researchers recently. Citizens of developed cities tend to proudly affiliate themselves to a particular city. They think that their city is unique and different. Those who live in New York, for example, think that nothing is like their city. They believe that their identity is different from those who live in London, Berlin, Warsaw or Tehran. So the place is an important factor in shaping people's identity. "Phenomenologically speaking, humans create a familiar and relatively controllable place for themselves" [6].

As shown before, local identity and civic pride can be considered one of the most important parameters in terms of discourse and identity of any place. To understand the mentioned parameters, one of the oldest cities in the Middle East was considered as a case study, namely Erbil/Hawler, Kurdistan Region, Iraq. Accordingly, the author of this study attempted to first study the Hawleri (it can be translated as Erbilian) identity. Hawleri is a local term used by Kurdish people to refer to people from Erbil which is the capital city of the Iraqi Kurdistan Region. Hawler is the local name of the city and "*i*" is a suffix that indicates the people of a certain place in the Kurdish language. Erbil (international term) is another name for Hawler which is used by people outside Kurdistan Region. In this study, both names are used interchangeably. Thus, this study is an attempt to examine the identity of Hawleri by studying the civic pride of the people, their experience and emotions, history, culture, beliefs and language variety.

Finally, it is important to show what is the novelty of this study and what is the reason to choose the oldest cities in the Middle East as a case study (i-iv).

(i) Both civic pride and Local identity have not been extensively studied as main parameters in the previous scientific researches related to identity and discourse, namely in most of the developing countries (as shown in the background section - §2). (ii) This study is different from the previous studies shown in the literature in terms of methodology and dataset because the critical discourse analysis has been used to reveal the people's identity and civic pride of a

city in an unidentified region. (iii) Erbil (Hawler) city is thought to be one of the oldest cities in the Middle East which is continuously inhabited by people for a very long history [7]. As the capital city of the Iraqi Kurdistan Region, it has its own characteristics; its people have their own ways of living and identity. (iv) The city is witnessing a rapid urban and socio-political transformation since 2003. "Economic prosperity has allowed the city to accelerate its reconstruction and development enjoying more stability and a safer environment in comparison to other cities in Iraq [8].

## 2. Literature review

Many studies have been conducted to examine different aspects of urban and city life. Some of them focus on the emotions and feelings of people to the geographical place, city or location, [9, 10]. Other studies deal with the relationship between discourse analysis and identity [3, 11–14]. Some other references focused on civic pride and community identity and how people from different places show their belongings [5, 15]. There are other studies that are not related to the topic of this study. In fact, they indirectly discuss the issues related to city identity, such as the relationship between city identity and economics, capitalism, inequality and transformation, politics, architecture, time, modernity and globalization [2, 16, 17]. As mentioned in the introduction section, in terms of methodology and dataset, this study is different from the previous studies showed above due to the fact that the critical discourse analysis was used to reveal the civic pride and people's identity of a city in an unidentified region.

Philosophers have not only considered the civic pride and local identities issues as the main concern of discourse analysts, in fact, they also attempted to study the role of place in human life. They studied human being's tendency to attach themselves to certain places. They believe that "people typically try to create familiarity and safety" [6]. In Being and Time, Heidegger [16] searches for the constituents of human existence, among other things. These basic features of all humans he calls "existentialia", or existential [16]. Apart from the place, Philosophers also focus on how people try to attach themselves to other people living around them. One of the props of our existence is "being with" others. The relations we have with our fellow human beings show our personality and nature. Existential is not only related to place and people, but also with things. Heidegger uses the philosophical term of "being alongside" things, which means dealing with different kinds of tools and objects, [16].

Another dimension of place identity can be attitudes and beliefs. It includes experience, emotions and feelings. Civic pride and city identity are a belief when someone feels that he or she is part of a larger community. This feeling is aspirational. The inhabitants of this particular place tend to improve themselves to better serve their community. People from this particular community have a sense of certain shared experiences and emotions. In this sense, as defined by Collins [18], pride is a value that tends to generate certain life up to ideals or expectations. Contrary, failure to live up to these ideals or expectations can damage or afflict a person's pride, and in some cases lead to feelings of self-doubt and shame. This is not only related to the place but also to the awareness of the citizens due to the fact that the constitution of identity is not only marked by an inward turning to "place", but also the awareness of self and other [19].

In studying urban identity, beliefs and attitudes are equally important with places in shaping the people's identity. Southworth [17] reported that urban identity is the best-recognized parameter in representation beyond time and includes a deep understanding of the characteristics of buildings along with social, cultural, and spiritual contexts. On one hand, identity refers to social and cultural cohesion that relates to a "number of things for an urban area and the people that live and work there. On the other hand, identity relates to tangible and intangible heritage: buildings, history, and memories [11].

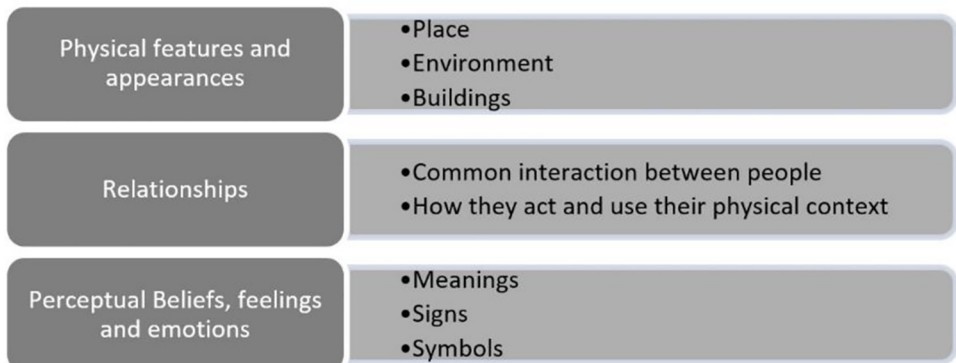

**Fig 1. Profile of the main urban identity aspects.**

Since the scope of the urban identity aspects is very wide, Fig 1 is drawn in order to simplify the information given in the literature. Accordingly, the urban identity aspects can be classified into three distinct features, namely (i) physical features and appearances, (ii) Relationships of people, and (iii) perceptual beliefs, feelings and emotions which are related to people's behaviors, intentions and experiences. Further details on the discourse analysis are shown in the next paragraphs.

Discourse analysis can be a suitable method to study civic pride and city identity. One of the fields in the discourse analysis domain is identity. Discourse analysts have interested in the way people show their belongings. Bethan and Elizabeth [20] reported that identity is actively, ongoingly and dynamically constituted in discourse. The authors believe that one of the purposes of discourse analysis is to explore identity forming in different discourse contexts. People in different settings use different discursive strategies, such as everyday dialogue (e.g. everyday interaction between friends), institutional discourse (e.g. government spokesmen briefings), narratives and stories (e.g. media stories), commodified contexts (e.g. business advertisements, spatial locations (e.g. neighborhood talks), and virtual settings (e.g. chat rooms).

In the analysis of civic pride discourse, some of the contexts and types of discourses previously mentioned can be useful, but the focus is more on spatial locations. This kind of study explores the discourse of material locations and spaces. As defined by Bethan and Elizabeth, [20], a basic idea is that "who we are" is intimately connected to "where we are", and that places can be moral sites of power struggle, exclusion and prejudice. People from different social and geographical contexts try to introduce themselves to one another, to which city, ethnicity, and social contexts they belong [13]. From different approaches of discourse analysis, critical discourse analysis is an appropriate approach to study identity construction and civic pride, since identity-forming is instilled in ideology and power relations which are included in the domain of critical discourse analysis [12].

Socially and Sociolinguisitcally, Hawleri is not a dialect by itself; it is part of a larger dialect that is spoken in a larger geographical region of Southern and Eastern Kurdistan. It is included with central Kurmanji which is spoken in many Kurdish populated cities of Iran and Iraq. Hawleri can be regarded as a variety of central Kurmanji [21]. Hawleri variety can be also regarded as a speech community, since this term is used to refer to people who use the same variety in the same geographical region and interact with each other using the same repertoire [14].

## 3. Methodology

In this study, the critical discourse analysis approach was used to analyze the data due to the fact that it is considered as a suitable approach to study civic pride and city identity according

to the literature. By using the tools of this approach, the Hawleri identity was examined in this study. Generally, the local identity of urban people can be studied using different methodologies, such as phenomenology, psychoanalysis and non-representational theory, but the focus of this study is on discourse analysis, namely critical discourse analysis due to the fact there is a special significance attached to the ways in which identity is realized in discourse in the selected approach that can be confirmed in a study [22]. According to the main concept of the selected approach, the questions (Q1-5) showed in Fig 2 are asked by Halwrei people.

To answer these questions, a large amount of data is gathered through the city social media channels (section 4.1), more specifically the Facebook group #Lo (لۆ# in Hawleri Kurdish) and randomly distributing a survey among a large number of people from Erbil (section 4.2). The data of the Facebook group are analyzed qualitatively and the survey responses are analyzed quantitatively. In order to avoid getting any random or unreliable sample (response), a strict boundary was considered to get the data. Thus, only those who answered all questions were considered from the online community (i.e. those who joined the Facebook group #Lo) and the survey (i.e. academic people). Further details regarding the dataset and the way how they were collected, namely for the online community and people from the survey are shown in sections 4.1 and 4.2, respectively. In general, 236 participants in the social media and 573 participants in the survey properly answered the questions showed in Fig 2, respectively. In addition, due to the strict boundary mentioned before, the percentage of people who concluded less

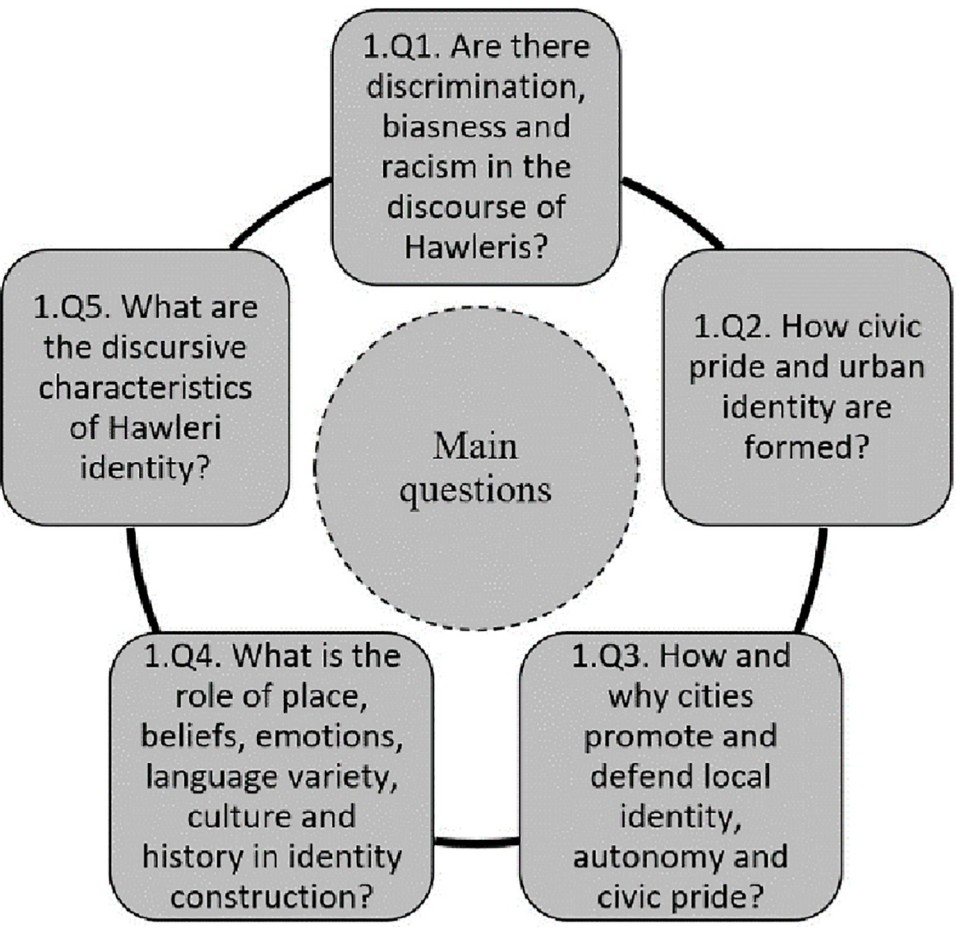

**Fig 2. The main questions asked from the Hawleri people.**

than bachelor degree is low (~9%). In other words, most of the selected answers came from academic people because they answered all questions. Apart from the mentioned fact, this study can be still representative of Erbil's population in general due to the fact that in discourse related matters (public options by leaders), many studies [23–25] concluded that the leaders and academic people have a significant impact on ordinary people (public option).

This study involved human participants and was reviewed and approved by the ethics committee of the Media Techniques Department, Erbil Technical Administrative College, Erbil Polytechnic University, Erbil, Iraq. With respect to the ethical consideration, the author assured the respondents with the cover letter about the anonymity with respect to their responses. Moreover, the author followed all ethical measure to complete the research. To answer the questions asked for the objective of this study, a large amount of data is gathered through the city social media channels and randomly distributing a survey among a large number of people from Erbil. For both groups, the questions were given after they expressed their willingness to participate as respondents of this research. Then, through a telephone call to inquire about the availability of all, before proceeding to send a survey through email. Due to respondent's comfortability and upon their request we obtained only verbal consents because they were not willing to provide written approval.

## 4. Results and analyses

### 4.1 Analyzing the results of the social media (online community)

As mentioned in the methodology section the Facebook group #Lo (#لۆ) in Hawleri Kurdish) is used to get the data of people's attitudes and beliefs. #Lo is a Facebook group supported by a page under the same title. The group has 77,252 members (at the time of conducting this study). It is formally a recognized page by Facebook. The name and the logo of the page are symbols that signify the identity of Hawleri and Hawler city. The name #Lo (#لۆ) is a word that means "why" in some contexts and "to a" in other contexts that is a sign of Hawleri sub-dialect (variety) of Central Kurmanji of Kurdish language. Furthermore, #Lo is widely used by Hawleri people, and in other dialects and varieties of Kurdish the word Bo (بۆ) is used instead.

The cover page contains two photos, merged into one, that again signify the identity of Hawler. They are the photos of Erbil citadel and Minaret. The citadel, is called Qelat or Qalay Hawler in Kurdish, is an ancient citadel located in the center of the city [26]. It has been recognized by the World Heritage List recently. According to UNESCO's database, the citadel of Erbil is a unique surviving heritage.

The written and iconographic records of this citadel indicate that its history goes to the times of Assyrians when Erbil was a political and religious center. This means that the citadel's history goes back to 7000 years ago or so. Erbil has an amazing continuity of its name (Irbilum, Urbilum, Urbel, Arbail, Arbira, Arbela and Erbil/Arbil) since pre-Sumerian times. The minaret is also located in Erbil. Its history goes back to the beginning of the 12ᵗʰ century. These two sites are signs of Hawler and Hawleri identity.

The policy of the group is stated by the chief admins in an introduction at the top of the page. The policy statement also contains numerous signs that indicate Hawleri identity. The title of the introduction says, "Let's make #Lo (#لۆ) the logo of Hawler", which is a clear indication of Hawleri identity. The content of the policy contains several sentences that signify the Hawleri identity:

• "Here is chief house (Diwaxan) of Hawleris";

• "Hawler is well-known for hospitality";

• "Our aim is to serve the Hawler and Hawleri language variety";

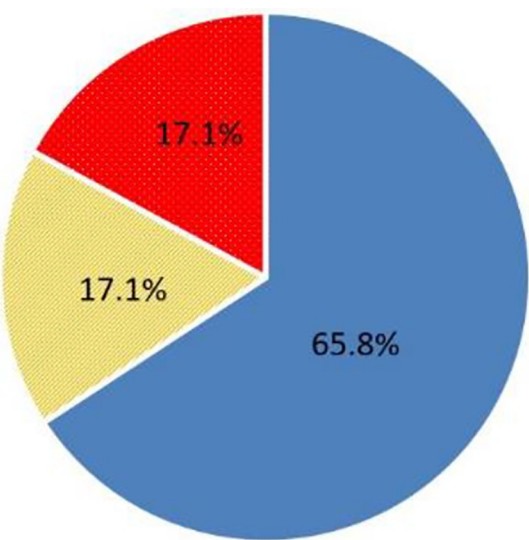

■ Hawleri is the person who lives in Erbil

※ Hawleri is a person who speaks Turkmani

■ Hawleri is a person who is Kurdish and his parents lived in Erbil

**Fig 3. The feedbacks of the social media group (#lo) members.**

- "Let's prove how our hearts are good and how Hawleri culture, variety and environment are high and beautiful".

This social media group can be regarded as an online community. In such kinds of communities, through the participation of people, social relations and issues of belonging and identity are signified. They feel as if they are in real communities just like a social club or a neighborhood team. The participants of this Facebook group #Lo have a lot of common things, such as, the same language variety which is Hawleri variety, the same geographical location and the same history and culture.

For the purpose of revealing the attitudes and stances of the members of this group, an article was posted on this Facebook group asking a question to be answered by the members. Basically, the question was "with regard to discourse and identity, linguistically and socially who is called Hawleri? 236 members answered and commented on the post. The answers and comments are classified and analyzed into three groups (I-III) as follows.

I. The majority of the answers state that geographical place is very important. They, 65.8%, believe that those who live within the bounds of Erbil city are Hawleris (Fig 3 - blue color);

II. A minor of 17.1% believe that only Turkmen and those who speak in Turkamni are regarded as Hawleris (Fig 3 - red color);

III. Another minor of 17.1% of the participants believe that original Kurds who live in the city, and those whose parents and grandparents lived in Erbil, are regarded Hawleris (Fig 3 - green color).

Fig 3 clearly shows the discourse of identity and how people try to show their ethnic and geographical belongings to the city. Looking for the hidden agenda in this kind of discourse one can find the hidden agenda and racism. The Kurds, who are the majority in the sample of the population, want to say that only Kurds are original Hawleris (17.1% of the total answers), whereas the Turkmen, who are the minority in the above sample, believe that only Turkmen are regarded as Hawleris (17.1% of the total answers). In addition to these two contradicting discourses, there is the third attitude (the majority– 65.8% of the total answers) which fortunately believe that those who live in the city, and those whose parents and grandparents lived in Erbil, are regarded Hawleris. Nevertheless, there is still a kind of discrimination in the latter discourse and attitude, since they think that a true Hawleri should not only himself live in Hawler, but their parents and grandparents should have been lived in Hawler.

## 4.2 The results of analyzing the survey

The following two tables (Tables 1 & 2) show the results of the survey. The survey is divided into two parts. The first part contains questions on the demographic information of the participants, such as age, residence information, birthplace, education and ethnicity, as shown in Table (1). The second part of the questionnaire contains questions on the aspects of Hawleri identity which are shown in Table (2).

As mentioned before, the survey was randomly distributed through email and social media channels. The demographic information shows that people of different ages have participated,

**Table 1. The demographic profile of the study's respondents.**

| Demographic | Characteristic | Frequency | Percentage |
|---|---|---|---|
| Age | Under 20 years | 14 | 2.4 |
| | 20–39 Years | 323 | 56.4 |
| | 40–60 year | 218 | 38 |
| | Above 60 years | 18 | 3.1 |
| Residence information | Erbil Center | 422 | 73.6 |
| | Suburban Erbil | 108 | 18.8 |
| | Outside of Erbil and other cities | 28 | 4.9 |
| | Outside Kurdistan Region | 15 | 2.6 |
| Birth Place | Erbil Center | 337 | 58.8 |
| | Suburban Erbil | 118 | 20.6 |
| | Outside of Erbil and other cities | 88 | 15.4 |
| | Outside Kurdistan Region | 30 | 5.2 |
| Parents' Birth Place | Parents born in Erbil | 271 | 47.3 |
| | Only father born in Erbil | 37 | 6.5 |
| | Only mother born in Erbil | 31 | 5.4 |
| | Parents not born in Erbil | 234 | 40.8 |
| Education Qualifications | Below primary School | 5 | 0.9 |
| | Preparatory School | 47 | 8.2 |
| | College & Institute graduates | 319 | 55.7 |
| | MA, MSc. and Ph.D. Holders | 202 | 35.3 |
| Ethnic Backgrounds | Kurds | 559 | 97.6 |
| | Turkmen | 9 | 1.6 |
| | Christians | 4 | 0.7 |
| | Others | None | None |

**Table 2. Halweri identity.**

| NO. | Descriptive questions | Characteristic | Frequency | Percentage |
|---|---|---|---|---|
| Q1 | Hawleri is someone who: | Who loves Erbil and has memories in the city | 140 | 24.4 |
| | | Who serves Erbil | 311 | 54.3 |
| | | Someone who thinks he/she is from Erbil | 122 | 21.3 |
| Q2 | Do you think you are Hawleri? | Yes | 507 | 88.5 |
| | | No | 44 | 7.7 |
| | | I don't know | 22 | 3.8 |
| Q3 | Why do you think you are Hawleri? | Because I was born in Erbil | 309 | 53.9 |
| | | Because I have lived here for a long time | 76 | 13.6 |
| | | I love Erbil and I am comfortable in the city | 186 | 32.5 |
| Q4 | If you are Hawleri, are you proud of that? | Yes | 502 | 87.6 |
| | | No | 17 | 3.0 |
| | | I don't know (maybe) | 54 | 9.4 |
| Q5 | Why are you proud of being Hawleri? | Because Erbil residents have many nice customs | 196 | 34.2 |
| | | Because I don't feel homesick in Erbil | 150 | 26.2 |
| | | Because Erbil has a glorious history | 227 | 39.6 |
| Q6 | In geographical terms, Hawleri is someone who. . . | Lived in Erbil Neighbourhoods (inside the 60-Meter Road) | 155 | 27.1 |
| | | Lived in Central Erbil (Inside the 120-Meter Road) | 191 | 33.3 |
| | | lives in Erbil Governorate from Makhmoor to Haji Omeran | 227 | 39.6 |
| Q7 | In terms of birth, Halweri is someone who . . . | Whose ancestors were born in Erbil | 270 | 47.1 |
| | | Was born in Erbil himself | 177 | 30.9 |
| | | Was born outside of Erbil, but he/she lives in Erbil | 126 | 22 |
| Q8 | In terms of language, Hawleri is someone who . . . | Speaks in Kurdish | 363 | 63.4 |
| | | Speaks Turkmani | 8 | 1.4 |
| | | Speaks Kurdish and Turkmani | 202 | 45.2 |
| Q9 | In terms of Dialects, Hawleri is someone who. . . | Speaks the dialect of Erbil | 330 | 57.6 |
| | | Speaks in Sorani—Slang Kurdish | 197 | 34.4 |
| | | Speaks in Kurdish combined with Turkish dialect | 46 | 8.0 |
| Q10 | What languages do you speak at home? | Kurdish | 558 | 97.4 |
| | | Turkmani | 15 | 2.6 |
| | | Other languages | 15 | 2.6 |
| Q11 | Do you prefer to use the Hawleri dialect when talking to other people not from Erbil | Yes | 356 | 62.1 |
| | | No | 83 | 14.5 |
| | | Sometimes | 134 | 23.4 |
| Q12 | When I use the Hawleri Dialect with other people not from Erbil, I use "Lo | Feel proud of myself | 445 | 79.4 |
| | | Feel shy | 9 | 1.6.0 |
| | | I don't know | 109 | 19.0 |
| Q13 | In public ceremonies. . . | I use the Hawleri Dialect | 263 | 45.9 |
| | | I speak the slang Kurdish | 300 | 52.4 |
| | | I speak foreign languages (Arabic, English, etc.) | 10 | 1.7 |
| Q14 | In terms of clothes, Hawleri is someone who | Wears Kurdish clothes | 104 | 18.2 |
| | | Wears formal clothes | 10 | 1.7 |
| | | Wears both types of clothes | 459 | 80.1 |

(*Continued*)

**Table 2.** (Continued)

| NO. | Descriptive questions | Characteristic | Frequency | Percentage |
|---|---|---|---|---|
| Q15 | Which of the following characteristics are found in Hawleri | Hospitality | 111 | 19.4 |
| | | Charity | 121 | 21.1 |
| | | Honesty | 53 | 9.2 |
| | | Integrity | 71 | 12.4 |
| | | Faithful | 63 | 11 |
| | | All the above characteristics | 154 | 26.9 |
| Q16 | If you are given the opportunity to live, where would you stay? | In Erbil | 482 | 84.1 |
| | | Other cities in Kurdistan Region | 69 | 12.0 |
| | | Overseas | 22 | 3.8 |
| Q17 | In historical terms, tens of years ago Erbil | Was a Kurdish city | 496 | 86.5 |
| | | Was a Turkmani city | 33 | 5.8 |
| | | Other ethnicities | 44 | 7.7 |
| Q18 | In old historical terms, thousands of years ago Erbil | Was a Kurdish city | 430 | 75.1 |
| | | Was a Turkmani city | 21 | 3.7 |
| | | Other ethnicities | 122 | 21.3 |

2.4% were under the age of twenty, 56.4% were twenty to forty, 38% were forty to sixty and 3.1% were above the age of sixty. What is important here is that the majority of the respondents, around 97%, are adults and they can rationally communicate their beliefs, attitudes, preferences and identities.

Regarding the residential information, 73.6% of the participants were from the center of Erbil, 18.8% were from sub-urban or outside the town of Erbil, 4.9% were from other cities and 2.6% were from outside Kurdistan Region. Concerning the place of birth, 58.8% were born in the center of Erbil town, 20.6% were born outside the town, 15.4% were born in other cities and 5.2% were born outside the Kurdistan region. Here, the data shows that the majority of the respondents are either born in the center of Erbil town or they live there.

Regarding the birthplace of the participants' parents, 47.3% of the participants' both father and mother were born in Erbil, 6.5% only their fathers were born in Erbil, 5.4% only their mothers were born in Erbil, and 40.8% their parents both were not born in Erbil or they were born in other cities. These are interesting data, since they reveal that city identity and civic pride is associated with the people themselves not their parents. Though more than half of the participants, 52.7%, their parents were not born in Erbil, still they regard themselves as Hawleris.

Concerning the participant's educational background, only 0.9% were illiterate or below primary school, 8.2% were high school graduates, 55.7% were graduates of university and 35.3 were masters and Ph.D. holders. Since the vast majority of the respondents, 99%, were literate and well-educated, the results of the survey can be reliable. Educated people may possibly communicate and understand the issues of civic pride and identity.

With regard to the participants' ethnic background, the vast majority, 97.6% were Kurds, 1.6% were Turkmen and 0.7 were Christians. This information is useful for later analysis of the factors of language and language variety, since the Kurdish language, especially Hawleri sub-dialect or variety is an indicator for Hawleri identity.

To examine the results of Table 2, the answers for each question (Q1-18) were analyzed in the following paragraphs.

Q1 is about who is regarded as Hawleri. The rate of 24.4% believes that Hawleri is someone who loves Hawler (Erbil) and has memories in the city, 54.3% believe that those who serve

Hawler are Hawleris and 21.3% believe that those who think that they are from Hawler are Hawleris. Here, according to the feedbacks and beliefs of the majority of the participants, 54.3%, those who serve the city are regarded as Hawleris.

Regarding Q2 (do you think you are a Hawleri?), the majority of the participants 88.5% answered "Yes", 7.7% said "No" and 3.8% said they "don't know". Again, this is related to the discourse of attitude and beliefs. Since they regard themselves as Hawleris, they are Hawleris.

Concerning Q3 (why do you think you are a Hawleri?), 53.9% said because they were born in Erbil, 13.6% said because they lived here for a long time and 32.5% said they love Hawler (Erbil) and they are comfortable in the city. This is related to emotions. Emotions have an influence on the identity of a person.

For Q4 (are you proud of being Hawleri?), 87.6% answered that they are proud of being Hawleris, 3% said no and 9.4% said they don't know. Again the emotion is clear here, because the majority of the participants feel proud of being Hawleris. In addition, Q5 is related to the previous question and also associated with emotions (why are you proud of being a Hawleri?), about 34.2% of participants said that they are proud of being a Hawleri because Hawleris have nice traditions, 26.2% said because they don't feel homesick in Hawler and 39.6% said because Hawler has a glorious history. These feelings and emotions are important in critical discourse analysis. This can be called the civic pride of Hawleri people, since they feel proud to be Hawleris.

Regarding Q6 which is on geographical factor, 27.1% of the participants believe that a Hawleri is someone who lived in the center of Erbil city (inside the 60-meter road), 33.3% consider that those who lived inside the 120-meter road are Hawleris and 39.6% believe that those who live inside the borders of Erbil governorate (from Makhmoor to Haji Omran districts) are Hawleris. The geographical factor is an important variable that signifies the identity of people.

Birthplace is another variable that is addressed in Q7. The ratio of 47.1% of the respondents stated that those whose ancestors were born in Erbil are Hawleris, 30.9% stated that those who were born in Erbil are regarded as Hawleris and 22% of those who born outside Hawler but live in Hawler are regarded as Hawleris. These results confirm the results of the demographic information shown in Fig 3, since 47.3% of the participants' both father and mother were born in Erbil.

Q8-13 are related to language, dialect, variety and their uses in different social settings. In terms of language, 63.4% of the participants stated that Hawleri people speak Kurdish, 1.4% believe that Hawleris speak Turkmani and 45.2% believe that Hawleris speak both Kurdish and Turkmani. In terms of dialect or variety, 57.6% of the participants believed that Hawleris speak the Hawleri Kurdish variety, 34.4% stated that Hawleris central Kurmanji (Sorani) dialect and 8% believed that Hawleris speak a combination of Kurdish and Turkmani. Regarding the language spoken at home, 97.4% of the participants spoke Kurdish at home, 2.6% spoke Turkmani and another 2.6% spoke other languages. These results reveal that Hawleri Kurdish variety of central Kurmanji (Sorani) is a widespread variety in Erbil and it is sign of Hawleri identity. In addition to Hawleri variety, Hawleris may speak the semi-standard Sorani and Turkmani.

Regarding the social status of Hawleri Kurdish variety (Q11), when the participants are asked whether they prefer to use Hawleri variety when talking to Non-Hawleris, 62.1% said yes, 14.5% said no and 23.4% said sometimes. This is an indication of having a good feeling or proudness of Hawleris, since they prefer to use their language variety even when communicating with non-Hawleris. In answering Q12 whether when they use the word lo (why or to) they feel proud or not, 79.4% said that "Yes" they feel proud, 1.6% said they feel shy and 19% said they don't know. Again this is related to feelings and emotions. Regarding question 13 which is about using the Hawleri variety in public ceremonies, 45.9% said they use it, 52.4 said they

use the Central Kurmanji (a sub-standard of Kurdish) and 1.7% stated that they use foreign languages in public ceremonies. All these results indicate that Hawleri variety has got a degree of social status, since Hawleris do not feel shy in communicating with others and they are proud of using it.

Q14 examines the discourse of clothes. The participants were asked whether Hawleris wear Kurdish traditional clothes or modern formal ones. 18.2% said that they wear Kurdish clothes, 1.7% said they wear modern clothes and 80.1% said they wear both types. Here, since Hawler is an urban city, the capital city of the Iraqi Kurdistan Region and it has become a metropolitan city, people of Hawler wear different kinds of clothes, traditional and modern.

Regarding Q15 which is about some good traits and virtues of Hawleris, 19.4% believe that hospitality is a virtue of Hawleris, 21.1% said that charity is a character of Hawleris, 9.2% stated that honesty is associated with Hawleri people, 12.4% believe that integrity is a feature of Hawleris, 11% said that faithfulness is a character of Hawleris and 26.9 believe that Hawleris have all the above merits.

Q16 asks the participants that if they are given the choice to live where they would stay. 84.1% stated that they prefer to live in Hawler, 12% said they prefer to live in the other cities of Kurdistan and 3.8% said that they would like to live abroad. This is also related to feelings and emotions, since the majority of the participants love to live in Hawler.

The last two questions (Q17&18), are concerned with history. In historical terms, 86.5% stated that hundreds of years ago Erbil was a Kurdish city, 5.8% said it was a Turkmani city and 7.7% believed that it was other ethnicities' city. In older historical terms, 75.1% believed that thousands of years ago Hawler was a Kurdish city, 3.7% stated that it was a Turkmani city and 21.3% believed that it was the other ethnicities city. Here, the national identity of Hawleri people is obvious. Since the majority of the population is Kurds, they believe that Hawler was a Kurdish city in the close and far history.

Apart from the above facts, further analysis must be done in order to understand the effect of the Ethnic Backgrounds on the Hawleri identity and show a correlation between study factors. Thus, the responses of the participants have been classified based on their Ethnic Backgrounds (Table 3). In addition, for each question, the most frequent response was selected and identified the percentage. The output of this analysis has been shown in Section 5.

Apart from the selected factors, the Physical properties of architectural identity is another parameter that must be considered. In other words, both moral and physical factors have a significant impact on identity matters. Nevertheless, these parameters have not been discussed in this study due to the fact that the author of this study has been studied the mentioned factors in the previous study [27].

## 5. Discussions

Discussing the analysis and results of the social media group comments and the survey, the study arrives at several arguments. In the discourse of civic pride and city identity, attitudes and beliefs are important. One can see that those who think that they are from Hawler are Hawleris. Another factor in civic identity is emotion. Emotions have an effective influence on the identity of a person. Those who feel, love and attach themselves to Hawler, they regard themselves Hawleris. Since they feel attached to the city, they are also proud of being Hawleris. Additionally, birthplace is another essential factor in identifying civic pride and identity. Those who were born in Erbil and their parents were also born in Erbil are clearly regarded as Hawleris. Geography is another factor. Those who lived in the center of Erbil are undoubtedly Hawleris, though those who live outside the city, within the boundaries of Erbil governorate are also regarded Hawleris.

**Table 3. Responses of the Descriptive questions based on the ethnic Backgrounds (Kurd, Turkmen and Christians).**

| NO. | Descriptive questions | Ethnic Backgrounds | | | | | |
|---|---|---|---|---|---|---|---|
| | | Turkmen | % | Christians | % | Kurd | % |
| Q1 | Hawleri is someone: | Who thinks he/she is from Erbil | 56 | Who serves Erbil | 75 | Who serves Erbil | 54 |
| Q2 | Do you think you are Hawleri? | Yes | 100 | yes | 100 | Yes | 88 |
| Q3 | Why do you think you are Hawleri? | Because I was born in Erbil | 89 | I love Erbil and I am comfortable in the city | 75 | Because I was born in Erbil | 53 |
| Q4 | If you are Hawleri, are you proud of that? | Yes | 100 | Yes | 100 | Yes | 88 |
| Q5 | Why are you proud of being Hawleri? | Because Erbil residents have many nice customs | 56 | Because I don't feel homesick in Erbil | 50 | Because Erbil has a glorious history | 39 |
| Q6 | In geographical terms, Hawleri is someone who... | Lived in Erbil Neighbourhoods (inside the 60 Meter Road) | 100 | Lived in Central Erbil (Inside the 120 Meter Road) | 50 | lives in Erbil Governorate from Makhmoor to Haji Omeran | 41 |
| Q7 | In terms of birth, Halweri is someone who ... | Whose ancestors were born in Erbil | 89 | Whose ancestors were born in Erbil | 50 | Whose ancestors were born in Erbil | 47 |
| Q8 | In terms of language, Hawleri is someone who ... | Speaks Kurdish and Turkmani | 67 | Speaks Kurdish and Turkmani | 50 | Speaks in Kurdish | 65 |
| Q9 | In terms of Dialects, Hawleri is someone who... | Speaks the dialect of Erbil | 56 | Speaks in Sorani—Slang Kurdish | 50 | Speaks the dialect of Erbil | 58 |
| Q10 | What languages do you speak at home? | Turkmani | 56 | Other languages | 100 | Kurdi | 97 |
| Q11 | Do you prefer to use the Hawleri dialect when talking to other people not from Erbil | Yes | 100 | Sometimes | 50 | Yes | 62 |
| Q12 | When I use the Hawleri Dialect with other people not from Erbil, I use "Lo | Feel proud of myself | 100 | I don't know | 75 | Feel proud of myself | 80 |
| Q13 | In public ceremonies... | I use the Hawleri Dialect | 56 | I use the Hawleri Dialect | 75 | I speak the slang Kurdish | 52 |
| Q14 | In terms of clothes, Hawleri is someone who | Wears Kurdish and formal clothes | 56 | Wears Kurdish and formal clothes | 100 | Wears Kurdish and formal clothes | 80 |
| Q15 | Which of the following characteristics are found in Hawleri | All (Hospitality, Charity, Honesty, Integrity, Faithful) | 89 | All (Hospitality, Charity, Honesty, Integrity, Faithful) | 50 | All (Hospitality, Charity, Honesty, Integrity, Faithful) | 64 |
| Q16 | If you are given the opportunity to live, where would you stay? | In Erbil | 100 | In Erbil | 100 | In Erbil | 84 |
| Q17 | In historical terms, tens of years ago Erbil | Was a Turkmani city | 67 | Other ethnicities | 50 | Was a Kurdish city | 88 |
| Q18 | In old historical terms, thousands of years ago Erbil | Was a Turkmani city | 44 | Other ethnicities | 50 | Was a Kurdish city | 76 |

The language variety is another variable. Hawleris speak a special variety which is included within central Kurmanji (Sorani dialect) of Kurdish. Hawleris have a repertoire of social belonging and discourse community membership recognized by their variety. They may try to use their language variety as a marker of closeness and group affiliation. Turkmani is the second language variety used by Turkmani Hawleris. Sometimes Turkmen tend to use Turkmani as a way to save their privacy and intimacy. As stated by Bloor and Bloor [22], people tend to identify themselves with their own social groupings (Self) and often place themselves in opposition to other social groupings (others). Moreover, Hawleris often use a semi-standard Kurdish of Central Kurmanji to communicate in formal contexts and to put themselves within a wider context of Kurdistan.

Through our use of language, we not only "display" who we are, but also how we want people to see "us". The way we dress, the gestures we use, and the ways we act and interact also influence how we display social identity. Other factors that influence this include the way we think, the attitudes we display, and the things we value, feel and believe. These facts are also confirmed in a study [14].

Discussing the civic pride and urban identity of Hawleris critically, one can find clues of racism, bias and discrimination. Starting from the answers of the last question of the survey on

the history of Erbil, there is a kind of excluding of other nationalities and ethnicities. For example, the participants did not mention the role of Assyrians (many historical sources indicate that Erbil was an Assyrian city thousands of years ago. At those times there were not national states and countries like Arabs, Kurds, Turks, etc. The participants' view can be regarded as a nationalistic view) in the history of the city. They just showed that Kurds are the original Hawleris in the city. While Assyrians lived there for thousands of years, since Hawler is a very ancient city in the Middle East. Some also marginalized the history of Turkmen too, who live in Hawler for hundreds of years. Regarding the characters of Hawleris, they attributed all the positive traits and virtues to Hawleris, as if Hawleris do not have any negative trait. Birthplace and geography are two areas that can lead to discrimination and prejudice. Some of the participants of the survey and in the Facebook group stated that those who were not born in Erbil are not regarded as Hawleris, not only that but some believed that for someone to be a Hawleri even their parents should be born in Erbil. This attitude excludes many decent families who have lived in Erbil for many decades.

According to the data shown in Table 3, the following facts can be discussed. The majority of Kurds and Christians believe that Hawleri is someone who serves Erbil, but Turkmens believes that Hawleri is the person who thinks he/she is from Erbil. Thus, Kurds and Christians judge based on the performance of the person while Turkmen cares about the way how the person believes (Table 3, Q1).

Only 88% of Kurdish participants believe that they are Hawleri and proud of it. Whereas All the Turkmen and Christian participants believe that they are Hawleri and proud of it (Table 3, Q2 and Q4). In addition, the majority of Kurds and Turkmen believe that they are Halwri because they are born in the city whereas Christians related the matter to the fact that they love Erbil and I am comfortable in the city (Table 3, Q3). In terms of the geographical, all Turkman participants believe that Hawleri is someone who Lived in Erbil Neighbourhoods (small area in the city center—inside the 60 Meter Road), and half of the Christian participants believe in the same matter except they slightly extended area (Inside the 120 Meter Road), whereas less than half of the Kurds believe that Hawleri is someone who Lived in Erbil region (bigger area—Erbil Governorate from Makhmoor to Haji Omeran) (Table 3, Q6).

In terms of birth, the ethnic background does not affect the response. The majority of them believe that Hawleri is someone whose ancestors were born in Erbil (Table 3, Q7). Nevertheless, in terms of the language they have different perspectives. The majority of Turkmen and Christians believe Hawleri is someone who speaks Kurdish and Turkmen whereas Kurdish participants believe Hawleri is someone who speaks Kurdish (Table 3, Q8). Furthermore, all the participants speak their language (Kurdish, Turkmani and other languages) at home (Table 3, Q10).

All Turkmen participants use the Hawleri dialect when talking to other people not from Erbil, whereas this percentage is less for Kurds and christen participants (Table 3, Q11). The same fact is true for Q12 (When I use the Hawleri Dialect with other people not from Erbil, I use "Lo). In terms of Q14 (clothes, Hawleri is someone who), Q15 (Which of the following characteristics are found in Hawleri) and Q16 (If you are given the opportunity to live, where would you stay?), the majority of the participants in all Ethnics are the same. In historical terms, None of the Ethnic (Turkmen, Christians, and Kurds) agree with each other. All of them believe that the city belongs to themselves (Table 3, Q17-18).

## 6. Conclusions

In this study, the discourse of civic pride and urban identity of the people are studied due to the fact that the knowledge on these two parameters is very scarce, namely for the developing

countries. Accordingly, one of the oldest cities in the Middle East was considered as a case study, namely Erbil/Hawler, Kurdistan region, Iraq. In order to implement a critical discourse analysis approach, the data were taken from a social media group of Hawleris and from the results of a survey distributed among the people of the city. Based on the feedbacks of the participants, the study has arrived at the following conclusions.

i.  Civic pride and local identity are found in the discourse of people around the world. In other words, the results of this study confirm the fact that local identity is a universal phenomenon. It is applicable to the people of all cities. They tend to show this kind of identity in many ways through different means. They show this identity in many ways through different means, such as, feelings and emotions, languages and local varieties, clothes and textiles, concentration on birthplace and geography and culture and history.

ii. Hawleri people love their city, speak a local variety of Kurdish language, wear Kurdish clothes with modern ones, they were born in Hawler or lived there for a long period of time and they share a common culture and history. They feel attached to the city and they are proud of being Hawleris.

iii. Critically analyzing the discourse of Hawleris, the strong feeling of Hawleriness sometimes leads to discrimination, biasness and racism. Hawleri Kurds sometimes exclude other nationalities and ethnic groups who live in the city. They are proud of themselves and they have only positive qualities and merits. They do not regard those who are not born in Hawler as Hawleris. Discrimination is also found in the discourse of Turkmani people; some of them believe that only Turkmen and those who speak Turkamni are regarded as Hawleris.

As a future study or to complement this work, the following questions can be considered for the survey: How does the author think that Hawleri identity competes with other forms of identity in Kurdistan? Does one feel Hawleri first and Kurdish second and a member of so and so tribe third and a Sunni (or Assyrian or something else) fourth, or some other order? Would a Kurdish Hawleri view a Turkmen Hawleri as closer to them than a Kurd from Suleimani?

## Author Contributions

**Conceptualization:** Kawa Abdulkareem Sherwani.

**Data curation:** Kawa Abdulkareem Sherwani.

**Formal analysis:** Kawa Abdulkareem Sherwani.

**Funding acquisition:** Kawa Abdulkareem Sherwani.

**Investigation:** Kawa Abdulkareem Sherwani.

**Methodology:** Kawa Abdulkareem Sherwani.

**Project administration:** Kawa Abdulkareem Sherwani.

**Resources:** Kawa Abdulkareem Sherwani.

**Software:** Kawa Abdulkareem Sherwani.

**Supervision:** Kawa Abdulkareem Sherwani.

**Validation:** Kawa Abdulkareem Sherwani.

**Visualization:** Kawa Abdulkareem Sherwani.

**Writing – original draft:** Kawa Abdulkareem Sherwani.

**Writing – review & editing:** Kawa Abdulkareem Sherwani.

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
