## [Decision Letter · Decision Letter 0]

7 Jan 2021

PONE-D-20-20350

The Discourse of Civic Pride: Hawleri Identity as one of the oldest Kurdistani cities in the Middle East

PLOS ONE

Dear Dr. Sherwani,

Thank you for submitting your manuscript to PLOS ONE. After careful consideration, we feel that it has merit but does not fully meet PLOS ONE’s publication criteria as it currently stands. Therefore, we invite you to submit a revised version of the manuscript that addresses the points raised during the review process.

The manuscript has been evaluated by two reviewers, and their comments are available below. You will see that the reviewers commented on the potential interest of your work. However, the reviewers have also raised critical concerns and the manuscript will need significant revision before it can be considered for publication. I have outlined some of the key concerns noted by the reviewers below, but you should respond all concerns mentioned by the reviewers in your response-to-reviewers document. 

The key concerns noted by the reviewers relate to external generalizability of the findings in light of the sampling strategy and the statistical analyses. These issues have limitations for the interpretation of the results and should be explored. Please note that novelty is not a requirement for publication in PLOS ONE: https://journals.plos.org/plosone/s/editorial-and-peer-review-process

We look forward to receiving your revised manuscript.

Kind regards,

Danielle Poole

Staff Editor

PLOS ONE

Journal Requirements:

3. Thank you for including your ethics statement:  "Erbil Polytechnic University, Erbil Technical Administrative College".  

Please amend your current ethics statement to confirm that your named institutional review board or ethics committee specifically approved this study.

6. We note that Figure 3 in your submission contain copyrighted images. All PLOS content is published under the Creative Commons Attribution License (CC BY 4.0), which means that the manuscript, images, and Supporting Information files will be freely available online, and any third party is permitted to access, download, copy, distribute, and use these materials in any way, even commercially, with proper attribution. For more information, see our copyright guidelines: http://journals.plos.org/plosone/s/licenses-and-copyright.

6.1.    You may seek permission from the original copyright holder of Figure 3 to publish the content specifically under the CC BY 4.0 license.

Reviewers' comments:

Reviewer's Responses to Questions

**Comments to the Author**

1. Is the manuscript technically sound, and do the data support the conclusions?

Reviewer #1: Yes

Reviewer #2: Yes

2. Has the statistical analysis been performed appropriately and rigorously? 

Reviewer #1: Yes

Reviewer #2: Yes

3. Have the authors made all data underlying the findings in their manuscript fully available?

Reviewer #1: Yes

Reviewer #2: Yes

4. Is the manuscript presented in an intelligible fashion and written in standard English?

Reviewer #1: Yes

Reviewer #2: Yes

5. Review Comments to the Author

Reviewer #1: 1) The writing is fine for the most part, but author needs to watch for typos and writing mistakes more. Too many appear for a work under review for publication.

2) The citation style here is one I’ve not seen before – is this the accepted reference style for this journal?

3) In the methodology section, author should probably comment a bit about how a question posed to a FB group does not result in a random sampling, since only those who choose to comment/answer the question, as well as join this group, have responses.

4) On the issue of education of respondents, some more needs to be said. With most of the respondents having a university education, it seems to me that they are not representative of Erbil’s population in general (most of whom do not have a university education).

5) P.22 – is it really justified from the data presented to say that Hawleris view themselves as “superior”?

6) The conclusion and analysis needs more. How does the author think that Hawleri identity competes with other forms of identity in Kurdistan? Does one feel Hawleri first and Kurdish second and a member of so and so tribe third and a Sunni (or Assyrian or something else) fourth, or some other order? Would a Kurdish Hawleri view a Turkmen Hawleri as closer to them than a Kurd from Suleimani? I know the data isn’t there to say much on the matter, but the author could speculate on such issues in the conclusion (“topics for further study….”). More analysis and explanation of the data itself is also needed in the main body of the paper.

Reviewer #2: The topic is interesting ... We think it will be so strong if the writer focused on Physical properties of architectural identity

The issue of architectural identity has become a common topic in architectural debates.

Based on literature from social science and humanities, this inquiry of “what is the identity” leads one to a question of personality (who are you?). The question of identity is often interpreted to be a question about people's concepts of “who they are” and how they relate to others…please apply both moral and physical factors on the issue of identity.

In term of statistical analysis, we think that the study need more improvement by adding Factor analysis, multi regression analysis or correlation between study factors…. depending on descriptive analysis will affect the originality of the study…

6. PLOS authors have the option to publish the peer review history of their article (what does this mean?). If published, this will include your full peer review and any attached files.

Reviewer #1: No

Reviewer #2: **Yes: **A.Prof.Dr.Salahaddin Yasin Baper

---

## [Decision Letter · Decision Letter 1]

18 Oct 2021

The Discourse of Civic Pride: Hawleri Identity as one of the oldest Kurdistani cities in the Middle East

PONE-D-20-20350R1

Dear Dr. Sherwani,

We’re pleased to inform you that your manuscript has been judged scientifically suitable for publication and will be formally accepted for publication once it meets all outstanding technical requirements.

Kind regards,

Avanti Dey, PhD

Staff Editor

PLOS ONE

Additional Editor Comments (optional):

Reviewers' comments:

Reviewer's Responses to Questions

**Comments to the Author**

1. If the authors have adequately addressed your comments raised in a previous round of review and you feel that this manuscript is now acceptable for publication, you may indicate that here to bypass the “Comments to the Author” section, enter your conflict of interest statement in the “Confidential to Editor” section, and submit your "Accept" recommendation.

Reviewer #2: All comments have been addressed

2. Is the manuscript technically sound, and do the data support the conclusions?

Reviewer #2: Yes

3. Has the statistical analysis been performed appropriately and rigorously? 

Reviewer #2: Yes

4. Have the authors made all data underlying the findings in their manuscript fully available?

Reviewer #2: Yes

5. Is the manuscript presented in an intelligible fashion and written in standard English?

Reviewer #2: Yes

6. Review Comments to the Author

Reviewer #2: (No Response)

7. PLOS authors have the option to publish the peer review history of their article (what does this mean?). If published, this will include your full peer review and any attached files.

Reviewer #2: **Yes: **A.prof.Dr. Salahaddin Yasin Baper

---

## [Editor Report · Acceptance letter]

27 Oct 2021

PONE-D-20-20350R1 

The Discourse of Civic Pride: Hawleri Identity as one of the oldest Kurdistani cities in the Middle East 

Dear Dr. Sherwani:

I'm pleased to inform you that your manuscript has been deemed suitable for publication in PLOS ONE. Congratulations! Your manuscript is now with our production department. 

Kind regards, 

on behalf of

Dr. Avanti Dey 

Staff Editor

PLOS ONE